# Development of MWCNT/Magnetite Flexible Triboelectric Sensors by Magnetic Patterning

**DOI:** 10.3390/polym15132870

**Published:** 2023-06-29

**Authors:** David Seixas Esteves, Manuel F. C. Pereira, Ana Ribeiro, Nelson Durães, Maria C. Paiva, Elsa W. Sequeiros

**Affiliations:** 1Department of Metallurgical and Materials Engineering, Faculty of Engineering, University of Porto, 4200-465 Porto, Portugal; ews@fe.up.pt; 2CENTI, Centre for Nanotechnology and Smart Materials, 4760-034 Vila Nova de Famalicão, Portugal; acribeiro@centi.pt (A.R.); nduraes@centi.pt (N.D.); 3CERENA, Center for Natural Resources and Environment, IST, University of Lisbon, 1049-001 Lisboa, Portugal; mfcp@tecnico.ulisboa.pt; 4Department of Polymer Engineering, Institute for Polymers and Composites, University of Minho, 4800-058 Guimarães, Portugal; mcpaiva@dep.uminho.pt; 5INEGI—Institute of Science and Innovation in Mechanical and Industrial Engineering, 4200-465 Porto, Portugal

**Keywords:** MWCNT, ferromagnetic, flexible and stretchable sensors, smart composites, sensor fabrication, polymer composites, polymer actuators, soft robotics

## Abstract

The fabrication of low-electrical-percolation-threshold polymer composites aims to reduce the weight fraction of the conductive nanomaterial necessary to achieve a given level of electrical resistivity of the composite. The present work aimed at preparing composites based on multiwalled carbon nanotubes (MWCNTs) and magnetite particles in a polyurethane (PU) matrix to study the effect on the electrical resistance of electrodes produced under magnetic fields. Composites with 1 wt.% of MWCNT, 1 wt.% of magnetite and combinations of both were prepared and analysed. The hybrid composites combined MWCNTs and magnetite at the weight ratios of 1:1; 1:1/6; 1:1/12; and 1:1/24. The results showed that MWCNTs were responsible for the electrical conductivity of the composites since the composites with 1 wt.% magnetite were non-conductive. Combining magnetite particles with MWCNTs reduces the electrical resistance of the composite. SQUID analysis showed that MWCNTs simultaneously exhibit ferromagnetism and diamagnetism, ferromagnetism being dominant at lower magnetic fields and diamagnetism being dominant at higher fields. Conversely, magnetite particles present a ferromagnetic response much stronger than MWCNTs. Finally, optical microscopy (OM) and X-ray micro computed tomography (micro CT) identified the interaction between particles and their location inside the composite. In conclusion, the combination of magnetite and MWCNTs in a polymer composite allows for the control of the location of these particles using an external magnetic field, decreasing the electrical resistance of the electrodes produced. By adding 1 wt.% of magnetite to 1 wt.% of MWCNT (1:1), the electric resistance of the composites decreased from 9 × 10^4^ to 5 × 10^3^ Ω. This approach significantly improved the reproducibility of the electrode’s fabrication process, enabling the development of a triboelectric sensor using a polyurethane (PU) composite and silicone rubber (SR). Finally, the method’s bearing was demonstrated by developing an automated robotic soft grip with tendon-driven actuation controlled by the triboelectric sensor. The results indicate that magnetic patterning is a versatile and low-cost approach to manufacturing sensors for soft robotics.

## 1. Introduction

Soft robotics is a new class of intelligent systems with specific properties that differentiate them from traditional rigid devices [1]. Instead of integrating rigid joints and links, where motion is predominantly localised at the joint, soft robots rely on continuum deformations to achieve motion. The control of these soft systems requires continuous monitoring of their state during action [1]. However, traditional sensors have high stiffness compared to the materials used in soft robotics [2,3]. Thus, developing a new class of soft sensors made from low-stiffness materials, such as those found in soft robots, is of tremendous interest. This new class of intelligent systems has the potential to be more adaptable and versatile, enabling robots to navigate complex environments and interact with humans and objects more safely and naturally. The potential applications of soft robotics are broad and range from medical devices to search and rescue operations, industrial automation and even space exploration. Their main advantages compared with traditional rigid devices are related to their flexibility, adaptability, durability and light weight [4,5,6]. 

The integration of flexible soft sensors can be particularly useful in soft robotic grippers, an area that has been growing considerably over time and is expected to grow even more by 2030 [7]. Soft robotic grippers significantly impact the close interaction between robots and their surroundings [8]. Safe and dexterous grasping is one of the main requisites for a wider implementation, but one of the major challenges is the accurate control of soft bodies. Therefore, a significant effort in perception has been made with embedded sensory systems [9]. 

The types of sensors used in soft robotic grippers are mostly proprioceptive sensors (e.g., Hall-effect sensors, encoders, torque sensors, tendon tension sensors, strain sensors) to estimate the position and velocity of the gripper elements or exteroceptive sensors (e.g., piezoelectric sensors, triboelectric sensors, resistive sensors, electromagnetic sensors) to gather information about external objects [10,11,12]. 

Regarding the exteroceptive sensors, the triboelectric sensors have been showing great potential due to their flexibility, material compatibility, and simplicity. Triboelectric sensors generate electrical signals through frictional contact between two dissimilar materials. The interaction between the generated charges is measured through the electrodes and can be used to sense motion, touch or pressure [13,14,15].

Like triboelectric sensors, many flexible soft robotics sensors require electrodes, which can be made using various conductive materials such as fillers [3,16,17]. MWCNTs have been used as effective fillers in polymer composites because of their high aspect ratio, low density, excellent mechanical strength and high thermal and electrical conductivities [18]. 

In a previous work [19], the authors reported the development of sensors that were compatible with soft robotics materials. The sensors presented low electrical resistance, using MWCNTs as conductive material and a magnetic patterning technique/method for fabricating electrodes. This magnetic patterning technique has the advantage of decreasing the weight percentage of MWCNTs needed to achieve the same electrical resistance compared to randomly aligned composites [18,20]. However, when the viscosity of the composites increases significantly due to the polymer cross-linking process, the MWCNTs’ low magnetic properties (assigned to the residual catalyst remaining in the MWCNT) decrease the quality of the magnetic patterning [21]. The solution to this issue was achieved by using magnetite particles (Fe_3_O_4_) combined with MWCNTs. The combination of MWCNT/magnetite has been introduced previously, typically involving the synthesis of Fe_3_O_4_/MWCNT by chemical processes [21,22,23,24,25,26,27]. As an example, Dong et al. [28] synthesised MWCNT/Fe_3_O_4_ to increase the electrical and dielectric properties of polyimide (PI) films. The chemical synthesis involves extra steps that can be avoided by physical mixing, facilitating sensor fabrication for soft robotics. 

This study explores the fabrication of triboelectric sensors for soft robotics applications, using the physical mixing of MWCNTs and magnetite to decrease the electrical resistance of the sensor’s electrodes and facilitate magnetic patterning. 

## 2. Materials and Methods

The presented work can be divided into three parts: materials, composites and prototype. The study of the materials consisted of evaluating three relevant aspects for the application: (i) the effect of the addition of particles and nanoparticles on the viscosity of the reactive polyurethane rubber; (ii) the differences in the characteristic magnetic properties of MWCNT and magnetite particles; and (iii) the interaction between the particles in a thermoset PU matrix composite and a static magnetic field applied to the composite during the cross-linking process. The second part involved characterising the magnetic patterns generated on the composites after cross-linking. Optical microscopy and X-ray computed microtomography were used to explore the particles’ distribution in detail. The ratio of particles with the electrical resistance of the fabricated electrodes was correlated, and the triboelectric sensors were characterised. Finally, a soft robotic prototype was developed in the last part, consisting of two tendon-driven actuators controlled by a triboelectric sensor.

### 2.1. Materials and Characterisation

The MWCNTs were supplied by Nanocyl (NC7000, Sambreville, Belgium). The synthetic magnetite powder, with an average particle dimension of 200 nm and an average density of 4.6 g/cm^3^, was supplied by Inoxia (Cranleigh, United Kingdom). The reactive PU rubber and the curing agent (Poly GlassRub 50—Transparent Polyurethane Rubber) were provided by Feroca (Madrid, Spain). The silicone rubber (Ecoflex 00-50) was supplied by Smooth-on, Inc. (Macungie, PA, USA). The neodymium N42 magnets with 0.5 × 0.5 × 0.5 cm^3^ were bought at K&J magnetics (Pipersville, PA, USA).

Regarding the characterisation processes, the PU viscosity was measured using a rotational viscometer (Fungilab-Master series smart, Barcelona, Spain) with an L2 spindle. The measurement was carried out for 30 min for each sample (PU without any additives, 1 wt.% MWCNT in PU and 1 wt.% MWCNT plus 1 wt.% of magnetite in PU). 

The magnetic properties of the MWCNT and magnetite powder were analysed using a superconducting quantum interference device (SQUID) magnetometer (Quantum Design, IFIMUP-IN). The SQUID magnetometers typically allow a fully automated measurement of the magnetisation of a specimen as a function of the magnetic field; the magnetisation as a function of the applied magnetic field (M(H)) was performed at 300 K for a maximum applied magnetic field of 50 kOe. OM analysis using a digital microscope (Leica DM 2500M, Wetzlar, Germany) characterised the particles and the permanent magnets. The variation of the magnetic response with PU curing time was assessed 20 min before and after PU cross-linking under a static magnetic field on composites containing 1 wt.% of MWCNT and 1 wt.% MWCNT + 1 wt.% of magnetite.

Finally, to ensure uniformity across the electrodes, the polarities of the magnets should be symmetrical. Therefore, to characterise the magnetic flux density of the magnets, a 3D magnetic characterisation was performed using a portable magnetic field mapper (M3D-2A-Port, Senis, Canton Zug, Switzerland). The tested area was 30 × 30 mm, and the distance between the permanent magnets and the probe was approximately 1 mm.

### 2.2. Composite Fabrication and Characterisation

After characterising the interaction between the particles and the permanent magnets, six different composites consisting of MWCNT + magnetite in a PU matrix were produced using the magnetic patterning process. Two magnet arrays arranged in an alternated polarity configuration generated the magnetic patterns. 

The fabrication process of the composites is illustrated in Figure 1. Six different composites were prepared. Two composites contained 1 wt.% of the individual particles: 1 wt.% MWCNT/PU and 1 wt.% magnetite/PU. The other four composites were prepared with different ratios of magnetite added to 1 wt.% MWCNT (1:1, 1:1/6, 1:1/12, 1:1/24). The different ratios of magnetite consisted of adding the extra weight percentage of magnetite to 1 wt% of MWCNTs. For example, for the sample (1:1), 1 wt.% magnetite is added to 1 wt.% MWCNTs, which means that the composite has a total of 2 wt.% of fillers. For the sample (1:1/24), 0.04 wt.% magnetite is added to 1 wt.% MWCNTs, which means that the composite has a total of 1.04 wt.% of fillers.

For the fabrication of the composites, the PU rubber was mixed with the particles by hand for 2 min; then, the mixture was poured into a mould (10 × 3 × 0.3 cm) and the mould was placed on top of the permanent magnet array. After 16 h, as described in the PU rubber datasheet, the specimen was de-moulded, and the process was repeated for each prepared sample and replicas. After de-moulding, a thin layer of PU was cut in both extremities, exposing the electrodes to test the electrical resistance of the electrodes. A thin silver ink (CI 1036, ECM, Delaware, OH, USA) layer was placed on each electrode and cured at 110 °C for 20 min. 

The morphological characterisation of the specimens was performed using a digital microscope (Leica DM 2500 M, Wetzlar, Germany). The images were acquired in transmission mode at different magnifications.

The micro CT was performed at the Laboratory of Mineralogy and Petrology of IST (LAMPIST-GEOLAB) to assess the composites’ three-dimensional (3D) microstructural and compositional heterogeneities. Digital radiographs were acquired on a micro CT SkyScan 1172 scanner (Bruker, Billerica, MA, USA) using an X-ray cone incident on a rotating specimen. Due to the specimens’ reduced size and composition, the experimental conditions were optimised for each specimen using a constant source power (10 W). The following operating conditions were used: source voltage of 60 kV, current of 165 µA, a pixel size resolution of 5–6 µm and an average of five radiographs per each position. The acquisition was performed by rotating the specimen over 180° with a 0.25° rotational step. Slice reconstructions were obtained with the NRecon 1.6.3 routine, and volumetric visualisation was achieved with DataView and CTvox programs, which integrate the instrument software packages. 

The electrical measurements consisted of testing the electrical resistance of each composite’s electrode. The dimensions of each electrode were approximately the dimensions of the magnet’s arrays (10 × 0.5 cm). The electrical resistance of the fabricated samples was measured by the two-point electrical resistance measurement technique (Keithley 6487, Cleveland, OH, USA). Each electrode of the six samples was tested. Four replicas of each sample were fabricated and tested to study the reproducibility of the fabrication process of the composites (1 wt.% magnetite, 1 wt.% MWCNT, 1:1, 1:1/6, 1:1/12 and 1:1/24). 

The triboelectric sensors were characterised using a high-speed digital multimeter (Keithley DMM7510 7 ½ Digital Multimeter, Cleveland, OH, USA), where both electrodes were connected to the multimeter terminals, and the output voltage was measured. Two materials were tested as friction layers, PU and SR. Both materials were pressed against the electrodes several times under controlled conditions in a three-point bending setup. The three-point bending tests were conducted on a Shimadzu AGX-V (Shimadzu, Kyoto, Japan) at 500 mm/min speed with an applied force of 10 N.

### 2.3. Prototype Development and Testing

The methodology for prototype development is presented in Figure 2. Regarding the sensor and actuator fabrication, first, the triboelectric sensor was patterned using 1 wt.% MWCNTs plus 1 wt.% of magnetite in PU. Then, the triboelectric sensor was placed inside the mould of the actuator and, finally, it was filled with PU rubber. The actuator’s mould consisted of a polymeric part produced by 3D printing. The structure also comprises two channels on the sides, where the string moves. The motion is comparable to the movement of a tendon, bending the structure after pulling the sting. The mould presents several folds on the front and back, facilitating the movement of the actuator and decreasing the force needed for the micro servo to pull the strings.

The mould and the prototype were designed with SolidWorks and fabricated using a 3D printing machine (Stratasys Fortus 250mc, Edina, MN, EUA). The hardware to control the prototype consisted of an Arduino Uno, electrical resistances, LEDs and two micro servo motors (DFRobot SER0006, Hubai, China). All the components were assembled inside the 3D printing prototype structure to test the prototype. 

## 3. Results and Discussion

### 3.1. Materials and Characterisation 

One of this work’s main objectives was to use magnetic patterning to fabricate electrodes to develop MWCNT/PU sensors. However, the pot life of the PU rubber was just 30 min, meaning that after mixing the two components the viscosity significantly increases, hampering the mixture’s workability and affecting patterning quality. Furthermore, the addition of MWCNTs to PU increases the viscosity, as shown in Figure 3, from approximately 4000 to 12,000 mPa or higher. Thus, magnetite particles were added to the composite to overcome the increase in viscosity and the low magnetic response of the MWCNTs. Increasing the magnetic response of the composite is expected to enhance the velocity of the particles inside the fluid, compensating for the increase in viscosity. Among the various magnetic additives available, magnetite was chosen for its attractive magnetic properties and low cost. 

The magnetic properties of the individual particles used to prepare the composite, magnetite and MWCNTs, were evaluated by SQUID magnetometry analysis. The SQUID analysis (Figure 4) shows that the MWCNTs present ferromagnetism and diamagnetism. This behaviour was explored in detail in previous work [19], where the magnetic response of the MWCNT was related to impurities from the fabrication process. Even though the MWCNT presents ferromagnetic properties, the values are relatively low compared to magnetite. The measured coercivity (Hc), retentivity (Mr) and saturation magnetisation (Ms) of the MWCNT at 300 K were 104.41 Oe, 0.13 emu/g and 1.07 emu/g, respectively. For the magnetite powder, the measured coercivity (Hc), retentivity (Mr) and saturation magnetisation (Ms) at 300 K were 64.01 Oe, 4.61 emu/g and 83.68 emu/g, respectively. Since magnetite has higher magnetic susceptibility than MWCNT, the magnetic attraction between the permanent magnets and the magnetite particles will also be higher [29]. A higher magnetic susceptibility can result in faster movement of the particles towards the permanent magnets inside fluids of similar viscosity. Therefore, when MWCNT and magnetite particles are mixed in a solution, the magnetic patterning quality is expected to increase under a static magnetic field [21,29]. According to McCloskey et al. [29], the magnetophoretic mobility in magnetic separation depends on inherent characteristics of the magnetic particles and the medium, such as medium viscosity, particle size and magnetic susceptibility of both the medium and the magnetic particles. There are also various forces influencing particle movement in a liquid suspension, including magnetic forces, buoyancy, gravity and drag forces.

The interaction between the particles (MWCNT and magnetite particles) in a PU solution was assessed by OM analysis, carried out before and after 20 min of the cross-linking process under a static magnetic field (Figure 5). A comparison of Figure 5a,b (and the videos provided in the Appendix A) shows that, in the case of the composite with 1 wt.% MWCNT + 1 wt.% of magnetite, there is an alignment of the particles with the magnetic field lines, creating a “magnetic mesh/net” that drags the MWCNT in the direction of the permanent magnet. However, the composite with MWCNT only shows that the MWCNT are dragged in the direction of the permanent magnet without any perceptive alignment. 

Figure 6 presents the magnetic characterisation performed on the permanent magnets array (Figure 6a) to produce the magnetic patterns/electrodes. The magnets show a configuration of alternated polarity (Figure 6b) and a magnetic flux density of approximately 260 mT for less than 1 mm between the probe and the magnet’s surface. Figure 6c shows that the magnetic flux density between the north and south poles is quite symmetric since both poles present values around 260 mT, guaranteeing a good uniformity of magnetic field across the location of the electrodes.

### 3.2. Composite Fabrication and Characterisation 

Figure 7 shows the longitudinal images of the magnet patterns (Figure 6a) created on the produced composite samples. A comparison of Figure 7 and Figure 6c shows a good match of the patterns formed with the applied magnetic field. The major difference between the three composites is the square patterns formed on 1 wt.% magnetite (Figure 7b), which is related to the higher magnetic flux density on the edges of the magnets [19].

The longitudinal view of the samples presented in Figure 7 depicts the formation of two parallel patterns along the composites, forming the electrodes. The composite with 1 wt.% MWCNT (Figure 7a) differs from the other specimens (Figure 7b,c), showing the presence of agglomerates of MWCNT between the parallel patterns, as well as apparent porosity inside the electrodes. These differences can be attributed to the small magnetic susceptibility of the MWCNT (Figure 4) and the increase in viscosity over time (Figure 3), hampering the movement towards the magnets. The composite with 1 wt.% magnetite (Figure 7b) is more transparent due to the fast displacement of the particles with high magnetic susceptibility towards the magnets. Finally, Figure 7c shows the sample (1 wt.% MWCNT + 1 wt.% magnetite) depicting two parallel patterns of conductive particles with significantly fewer MWCNT agglomerates between the electrodes and with higher MWCNT density along the electrodes. Thus, adding magnetite to MWCNT increases the composite’s overall magnetic properties, dragging the MWCNT along the magnetic field, increasing the pattern quality.

OM cross-section analysis of the composites is presented in Figure 8. The images depict the side view (across the thickness) of one electrode of each composite (1 wt.% MWCNT, 1 wt.% magnetite and (1 wt.% MWCNT + 1 wt.% magnetite)). Comparison of Figure 8a–c confirms that the MWCNT agglomerates do not align along the magnetic field. Instead, the low-magnetic-susceptibility MWCNT agglomerates are dragged toward the magnets. Conversely, the composite with 1 wt.% magnetite (Figure 8b) presents magnetic particles aligned in the magnetic field direction and particles accumulate on the location of the permanent magnets. The sample with 1 wt.% MWCNT + 1 wt.% magnetite (Figure 8c) presents a mixture of both behaviours. The magnetite particles align with the magnetic field, and small agglomerates of MWCNT are dragged toward the magnets.

The distribution of the magnetite particles, MWCNT agglomerates and the combination of both inside the composites was analysed by micro CT, with a particular focus on the region close to the permanent magnets (formation of the electrodes in the composite). The micro CT images (Figure 9) highlight the location of high-density particles inside the polymer composite. Since the density of the MWCNTs is slightly higher than the polymer, their location inside the matrix is difficult to identify (Figure 9a). In contrast, the density of magnetite particles is much higher than the polymer and MWCNTs, making them easier to localise in the composite (Figure 9b). The results from OM and micro CT could be compared, showing that the composite with 1 wt.% of magnetite presents a similar magnetic pattern in Figure 7b (in the vicinity of one of the square permanent magnets) and Figure 9b. Figure 9c depicts the composite formed by 1 wt.% MWCNT + 1 wt.% magnetite, showing the presence of the magnetite particles across the entire area close to the permanent magnet. The images show the distribution of magnetite particles and MWCNT agglomerates, indicating that their homogeneous mixture in the composite allowed a continuous distribution of both particles (MWCNT and magnetite particles) in the magnetic field generated by the permanent magnets. This continuous path of electrically conductive particles is expected to positively influence the quality of the electrodes formed in the composite.

The electrical resistance of each electrode formed in the composites was measured, and the results are presented in Figure 10. In this study it was confirmed that the electrical resistance measured for the composites with MWCNTs and magnetite is attributed mainly to the MWCNTs. When measuring composites with magnetite only, the electrodes were non-conductive, with an electrical resistance of (1.38 × 10^10^ ± 3.41 × 10^9^) Ω. On the contrary, when measuring composites with MWCNT only, the electrical resistance of each electrode was (8.85 × 10^4^ ± 3.09 × 10^4^) Ω. However, the interaction of the magnetite particles with the MWCNT agglomerates in the presence of a static magnetic field seems to increase the magnetic force exerted on the MWCNTs, generating a “magnetic net” that drags the particles in the direction of the permanent magnet. As shown in the OM analysis (Figure 7c) and micro CT images (Figure 9c), the mixture (MWCNTs plus magnetite particles) increases the concentration of conductive particles in the region of the static magnetic field, decreasing the electrical resistance of the electrode.

A comparison of the electrical resistance of the composites with different ratios of magnetite added to 1 wt.% of MWCNT is plotted in Figure 10. A decrease in the weight percentage of magnetite is observed to increase the electrical resistance of the composites, while keeping them electrically conductive. The average electrical resistance measured for the electrodes of the 1:1 composite (1 wt.% MWCNT + 1 wt.% magnetite) was (5.31 × 10^3^ ± 0.29 × 10^3^) Ω, while for the composite with 1 wt.% MWCNT + 0.04 wt.% magnetite (1:1/24) it increased to (29.2 × 10^3^ ± 10.1 × 10^3^) Ω. Therefore, a gradual decrease in the magnetite content causes a gradual increase in both the electrical resistance and its standard deviation. Even though the composite with 1 wt.% magnetite is nonconductive, the addition of even a small amount of magnetite to MWCNT significantly decreases the electrical resistance and reproducibility of the fabrication process. The increase in the standard deviation of electrical resistance as the concentration of magnetite decreases is probably related to a decrease in the effectiveness of the “magnetic net” formed between MWCNT and magnetite in the composite. Decreasing the concentration of magnetite will hamper the continuous and homogeneous drag of the MWCNT agglomerates towards the permanent magnets, decreasing the concentration of the electrically conductive particles in their vicinity.

After the fabrication of the conductive composites with different magnetite concentrations added to 1 wt.% MWCNT, the composites were tested as triboelectric sensors. Figure 11 presents the result of the tests performed on triboelectric sensors composed of different MWCNT and magnetite ratios (1:1, 1:1/6, 1:1/12 and 1:1/24) with two different friction layers on top (PU and SR). Figure 11a,b displays a positive impact of the decrease in electrical resistance of the electrodes on the output voltage for the two friction layers tested. An increase in the triboelectric effect was also observed when the composition of the friction layer is different from the matrix polymer. For the combination of PU as the friction layer and as the polymeric matrix, the output voltage should be close to 0 V since a flow of electrons between both materials should not exist during contact electrification. However, the fact that the PU samples still show some triboelectric effect could be associated with the interaction between the PU and the metallic parts of the tensile test machine tool under cyclic force [30].

The increase in the output voltage observed for the combination of PU with the SR may be interpreted as resulting from the tendency of the SR to attract electrons and the tendency of the PU to “give” electrons. When the surfaces of these materials contact each other, there is a flow of electrons that can be measured, and the opposite happens after the materials’ separation [31].

SR is a commonly used material in soft robotics applications, and, since it increases the output voltage of the triboelectric sensors, it was selected as the friction layer for the prototype.

### 3.3. Prototype Development and Testing 

After characterising the triboelectric sensor, the next step was the design of an actuator that could be produced based on a PU rubber matrix and controlled by an Arduino Uno. Among the different types of soft robotics actuators, the tendon-driven actuator was selected, which bends the polymeric structure when a string is pulled (Figure 2). For the fabrication of the prototype presented in Figure 12 and in (Appendix A), two actuators and two micro servo motors were required. Then, the Arduino Uno was used to measure the triboelectric signal and control the micro servo motors. The developed system works by measuring voltage changes on the triboelectric sensor. Since the Arduino does not measure negative output voltage, the triboelectric sensor was connected to a 3.3 V power supply. The system working principle can be interpreted as a sum of signals: a direct current (DC) signal with an alternating current (AC) signal whenever the sensor works. This way, when an object touches the sensor, there is a peak that almost achieves 0 V (Figure 12a). When the object is held by the actuator, there is no exchange of electrons between the materials, and the signal stays constant (Figure 12b). Finally, when the object is released from the actuator (Figure 12c), there is a positive peak achieving 5 V. By defining thresholds above and under 3.3 V, activating and deactivating the blue LED and the actuators was possible. The blue LED can be used to test the system without the actuators. The green LED is connected to the actuators, showing that the actuation system is on. 

The developed system successfully detects the presence of an object, closing its grip and grabbing it until it is pulled away from the gripper. When the system detects an opposite signal, the system automatically opens the grips. A system that closes and opens automatically increases user interaction, which is useful for demonstration purposes. However, the sensor can be used as a preventive measure to detect problems on the soft robotic gripper, sending alerts to the user in case of slippage or malfunction. 

## 4. Conclusions

This work reports the fabrication of low-cost triboelectric sensors using magnetic patterning. MWCNT and magnetite particles in a PU matrix formed the composites prepared for the magnetic patterning. During the curing of the PU, the MWCNT and magnetite particles were driven by a magnetic field to produce a pattern consisting of two parallel lines that formed the electrodes. These electrodes were used as triboelectric sensors combined with an SR friction layer. The developed sensors were integrated into a soft robotic actuator to demonstrate the applicability of the fabrication process.

The combination of magnetite with MWCNTs increased the magnetic interaction of the filler particles with the magnetic field generated by the permanent magnets, leading to the good quality of the magnetic patterning in a viscous thermoset matrix, producing electrodes with low electrical resistance. A decrease in the electrical resistance of the electrodes produced based on MWCNTs was observed with the addition of magnetite as compared to electrodes with MWCNT alone. The effect of magnetite was that of a “magnetic net” that effectively dragged the MWCNTs in the direction of the permanent magnets. This effect was illustrated by OM, observing the particle morphology along the PU cross-linking process, and by micro CT, which allowed the observation of the morphology of composites with 1 wt.% of MWCNT, 1 wt.% of magnetite and a mixture of 1 wt.% of each (1:1). The addition of 1 wt.% of magnetite to 1 wt.% of MWCNT in the PU composite decreased the electrical resistance of the electrodes formed from (88.5 × 10^3^ ± 30.9 × 10^3^) Ω to (5.31 × 10^3^ ± 2.89 × 10^2^) Ω. The effect of reducing the wt.% of magnetite added to MWCNTs was evaluated, showing that it increased the electrodes’ electrical resistance and decreased the fabrication process reproducibility. Finally, it was observed that the composites with lower electrical resistance presented higher triboelectric properties. The combination of PU and silicone rubber increased the output voltage under press and release motion, and this combination was used to develop a demonstration prototype. 

The prototype consisted of a soft robotic grip with a sensor and actuator that could detect the presence of an object and close the grip automatically. The system automatically opens when the object is pulled away from the grip. The developed tendon-driven actuators were activated by two micro servo motors controlled by the developed triboelectric sensor.

The magnetic patterning process developed proved to be an effective method to decrease the electrical resistance of conductive patterns. This simple, versatile and low-cost method can be used to develop soft robotics solutions to monitor the interaction between the soft robot and external objects. Even though the presented prototype was developed for proof of concept only, it is possible to programme it for other applications, such as slippage detection, for the already available soft robotics in the food industry. It could also be used to develop smart prosthetics, making movements and interaction with objects more natural.

## Figures and Tables

**Figure 1 polymers-15-02870-f001:**
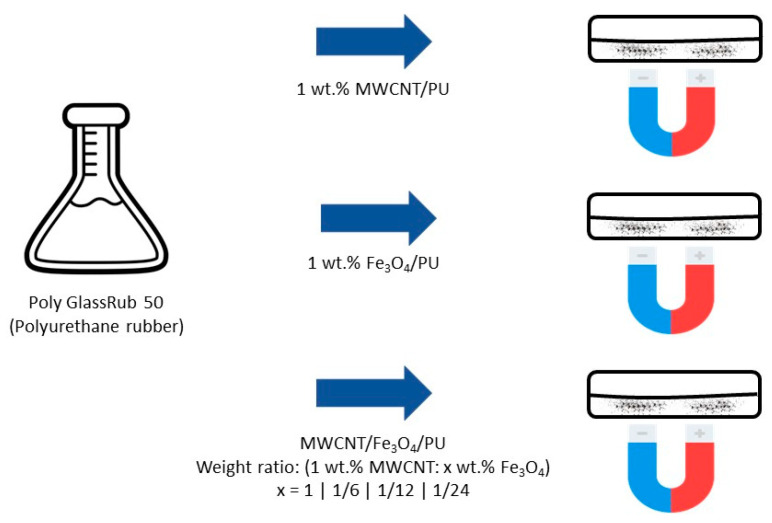
Composite materials fabrication process.

**Figure 2 polymers-15-02870-f002:**
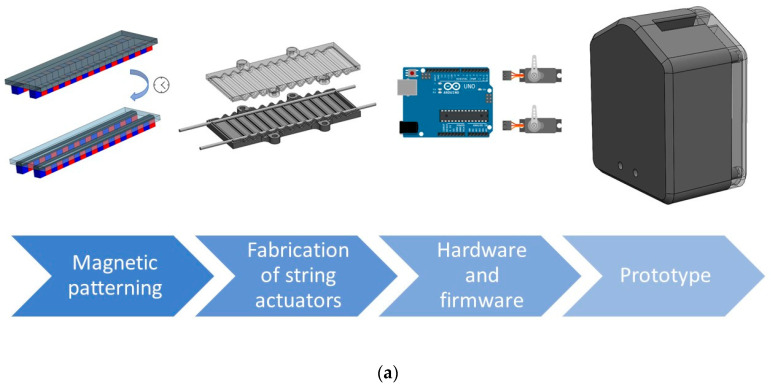
(**a**) Methodology for the prototype development and (**b**) development of a tendon-driven actuator with an integrated triboelectric sensor.

**Figure 3 polymers-15-02870-f003:**
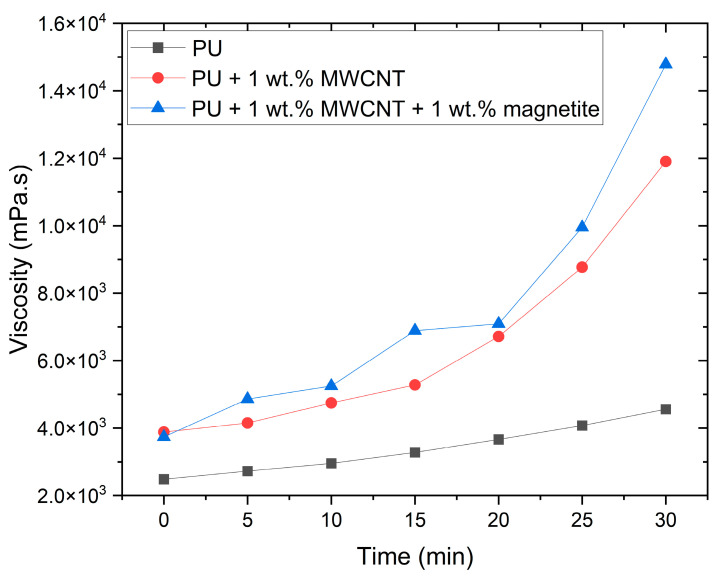
Characterisation of the viscosity of the PU thermoset over time. The viscosity was analysed for pure PU and PU composites.

**Figure 4 polymers-15-02870-f004:**
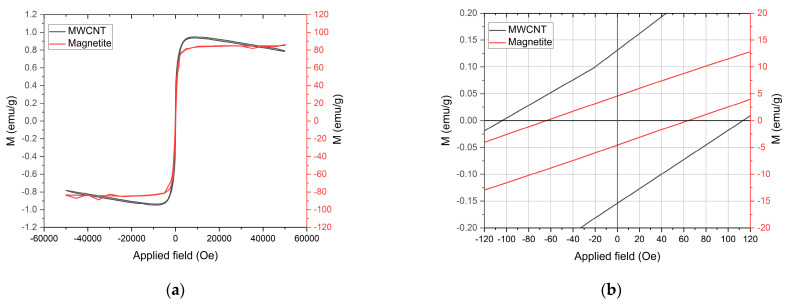
SQUID analysis of the MWCNT and magnetite particles. (**a**) From −50,000 to 50,000 Oe. (**b**) From −120 to 120 Oe.

**Figure 5 polymers-15-02870-f005:**
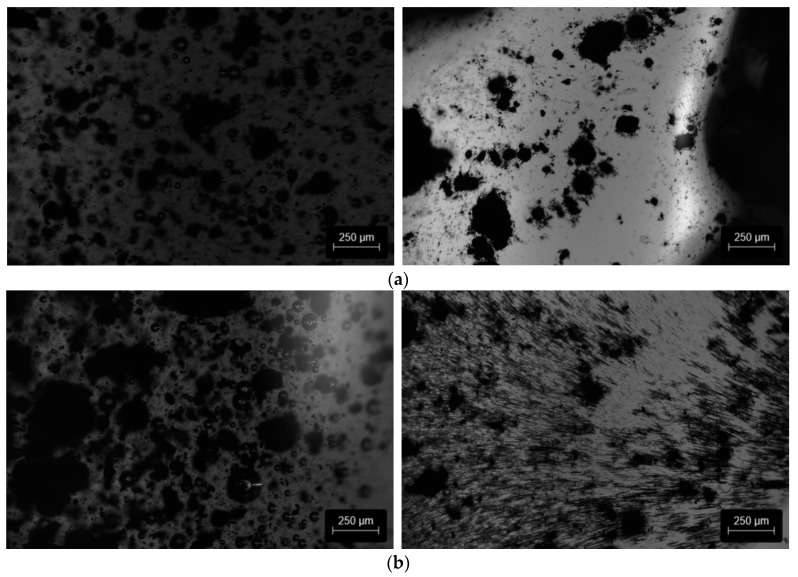
OM analysis of the organisation of the particles over time under a static magnetic field. (**a**) Composite with 1 wt.% of MWCNT and (**b**) 1 wt.% MWCNT + 1 wt.% of magnetite. The images on the left were acquired before cross-linking and on the right 20 min after the cross-linking process started, under the applied magnetic field.

**Figure 6 polymers-15-02870-f006:**
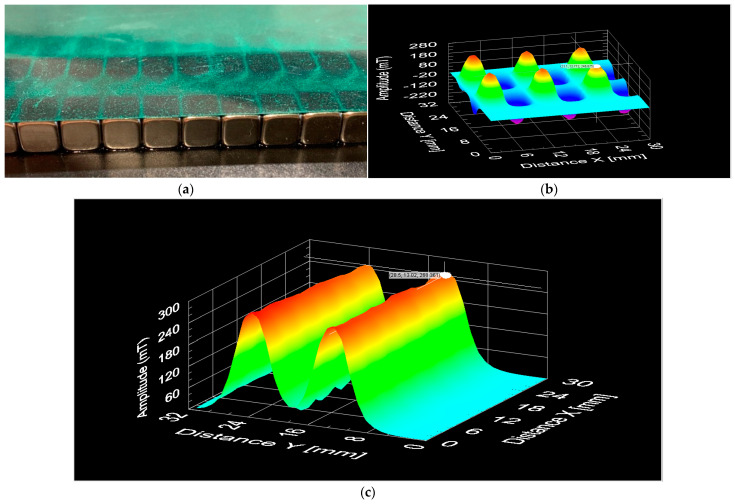
Magnetic characterisation of the permanent magnets used for the magnetic patterning of the electrodes. (**a**) Permanent magnets with a magnetic film. (**b**) Three-dimensional north pole and south pole distribution in the Z axis. (**c**) Three-dimensional symmetry analysis for the north and south poles in the Z axis.

**Figure 7 polymers-15-02870-f007:**
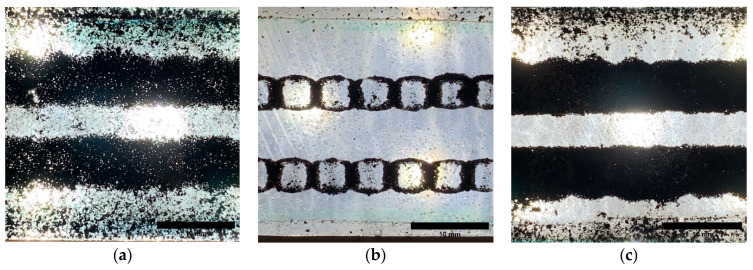
Longitudinal images of the magnetic patterns created on the composite samples produced. (**a**) Composite with 1 wt.% MWCNT/PU, (**b**) 1 wt.% magnetite/PU and (**c**) (1 wt.% MWCNT + 1 wt.% magnetite)/PU. (OM, transmission).

**Figure 8 polymers-15-02870-f008:**
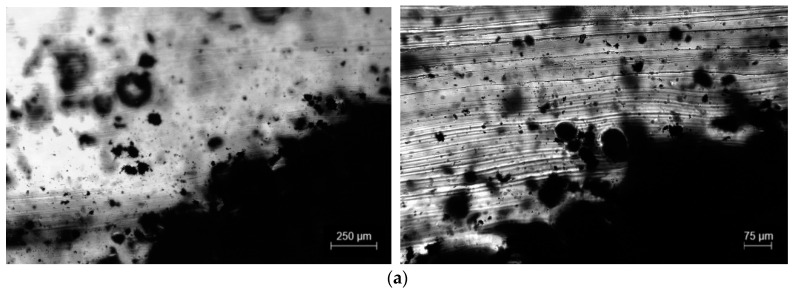
Cross-section images of the electrodes obtained by OM in transmission: (**a**) 1 wt.% MWCNT/PU, (**b**) 1 wt.% magnetite/PU and (**c**) (1 wt.% MWCNT + 1 wt.% magnetite)/PU.

**Figure 9 polymers-15-02870-f009:**
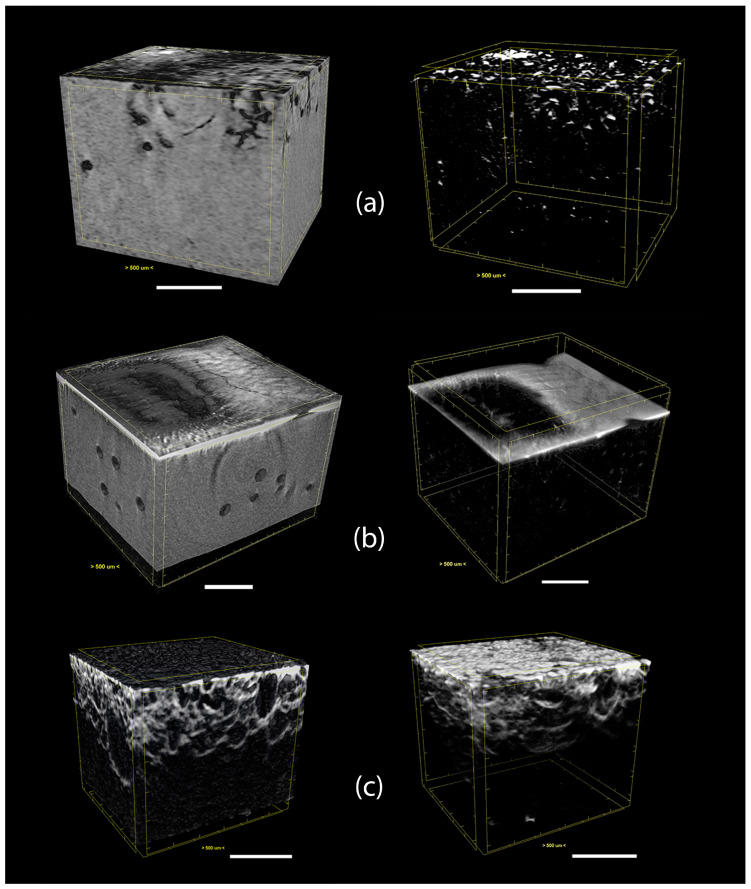
Micro CT analysis of the particle distribution inside the PU matrix for (**a**) 1 wt.% MWCNT/PU, (**b**) 1 wt.% magnetite/PU and (**c**) (1 wt.% MWCNT + 1 wt.% magnetite)/PU. The images on the left present the polymeric matrix and the particles, and the images on the right present just the particles.

**Figure 10 polymers-15-02870-f010:**
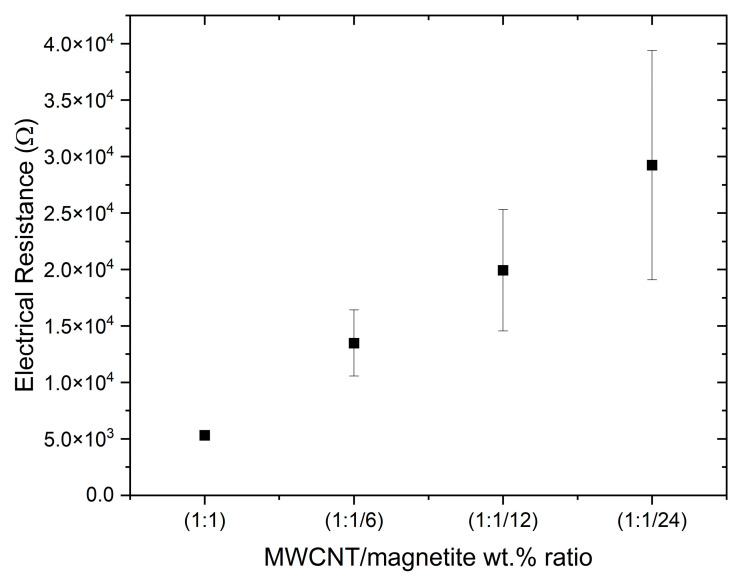
Electrical resistance of each electrode for composites with different ratios of magnetite added to 1 wt.% of MWCNT/PU.

**Figure 11 polymers-15-02870-f011:**
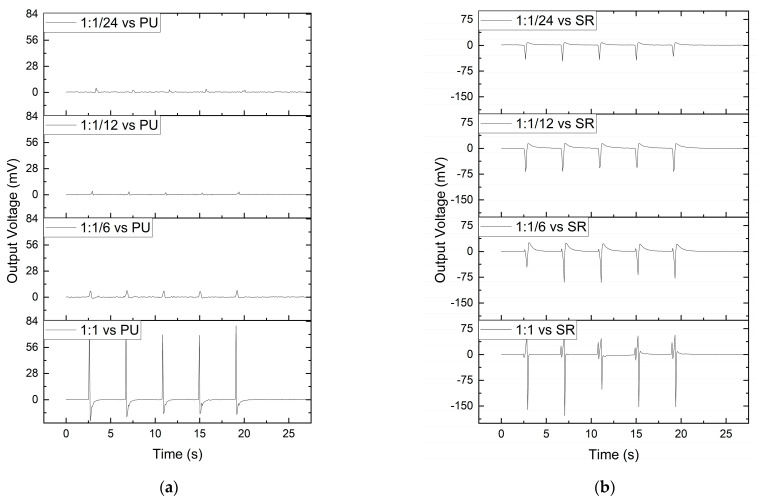
Triboelectric sensor characterisation with (**a**) a PU layer on top, compared with (**b**) an SR layer on top.

**Figure 12 polymers-15-02870-f012:**
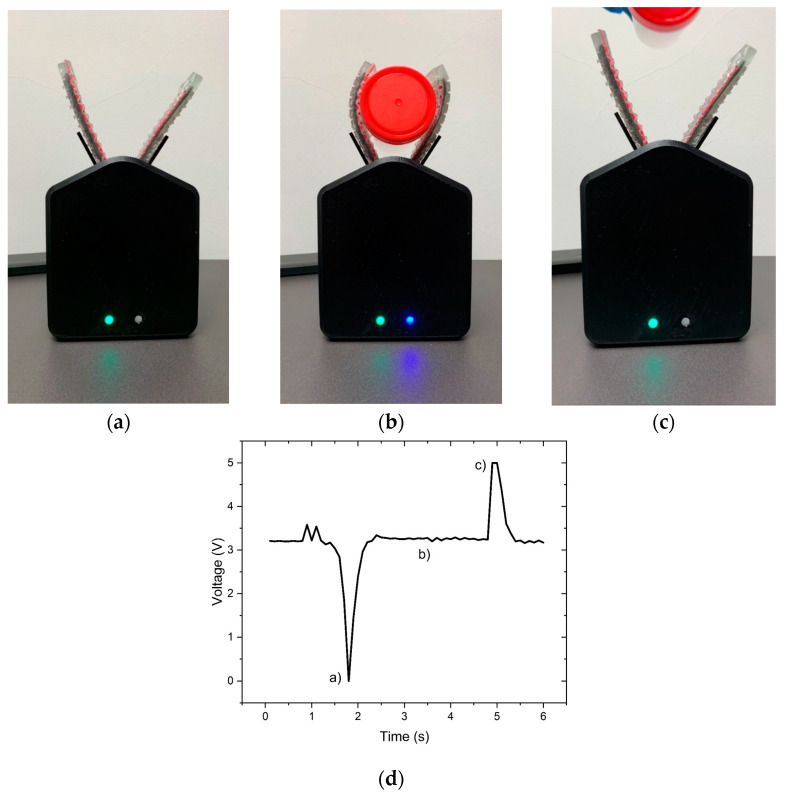
Soft robotics prototype using a triboelectric sensor (**a**) catching, (**b**) holding and (**c**) releasing an object when pulled away. (**d**) Output voltage signal at the three different stages (a) catching, (b) holding and (c) releasing an object when pulled away.

## Data Availability

Not applicable.

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
