# Peer review of "Development of MWCNT/Magnetite Flexible Triboelectric Sensors by Magnetic Patterning"

_polymers, 2023, doi:10.3390/polym15132870_

Round 1

Reviewer 1 Report

Esteves et al develops smart composites based on MWCNT or magnetic fillers, PU or Ecoflex for fabricating flexible triboelectric sensors. Interesting results are reported, and the concept is clearly and well-presented  However, the paper need some revisions before acceptance. Few points are –

[1] Many places in paper, its written “Error! Reference source not found”. Please delete or provide suitable reference.

[2] All the figures are Fizzy. Please provide high resolution images.

[3] In Figure 4b, please increase the number of data points to obtain a smooth magnetic behavior.

[4] From optical images, the tube-shaped morphology of the MWCNT is not visible. So, please provide high resolution image from SEM or TEM and discuss its dispersion.

[5] Ecoflex is the commercial name of silicone rubber (SR). So, please replace Ecoflex with SR in whole paper.

Good Luck for revisions!

Author Response

Reviewer 1

Esteves et al develops smart composites based on MWCNT or magnetic fillers, PU or Ecoflex for fabricating flexible triboelectric sensors. Interesting results are reported, and the concept is clearly and well-presented  However, the paper need some revisions before acceptance. Few points are –

Dear Reviewer, we appreciate your time reading and providing feedback on our manuscript.

[1] Many places in paper, its written “Error! Reference source not found”. Please delete or provide suitable reference.

You are right. There was a formatting problem when the document’s template was changed. All the figures are linked with the text now.

[2] All the figures are Fizzy. Please provide high resolution images.

Thank you for your suggestion. The quality of the figures was improved to 600 dpi.

[3] In Figure 4b, please increase the number of data points to obtain a smooth magnetic behavior.

New tests must be performed to increase the number of data points, which would take time for us (nearly a month due to being an external service with low availability). However, to increase the quality of the figure, we changed Figure 4b to focus on the most important part of the graph, the regions of the graph that cross the x=0 and y=0 lines. Therefore, we included the data from -120 to 120 Oe, facilitating the analysis of the values of magnetic coercivity (Hc) and retentivity (Mr) of the particles. The magnetic saturation of the particles can be seen in Figure 10 a).

[4] From optical images, the tube-shaped morphology of the MWCNT is not visible. So, please provide high resolution image from SEM or TEM and discuss its dispersion.

We did some SEM trials to identify the magnetite’s location inside the polymeric matrix using SEM-EDS. However, since it is an external service the SEM specialist could not take high-quality images, allegedly due to the accumulation of charges on the polymeric matrix. Since it was impossible to find the magnetite's location using SEM-EDS we decided to do micro CT characterization. Since the magnetite has higher density than both the polymer and the MWCNT it was possible to find its location inside the composite. From the results, it was possible to understand that for the sample with 1 wt.% of magnetite, the particles accumulated on the edge of the permanent magnets. Conversely, when adding 1 wt. % of magnetite to 1 wt.% of MWCNT, the magnetite is well mixed with the MWCNT.

Regarding the analysis of the tubular shape of the MWCNT, we also did some trials with the MWCNT powder, but as we explained in the paper, the MWCNT are agglomerated due to the strong Van der Waals forces (images below).

Since the focus of this work was not the dispersion of the MWCNT, because it would affect the magnetic pattern quality, we decided not to include these images.

[5] Ecoflex is the commercial name of silicone rubber (SR). So, please replace Ecoflex with SR in whole paper.

Thanks for the suggestion, Ecoflex is now replaced with SR along the text.

Good Luck for revisions!

Dear Reviewer, we are confident we have addressed all your concerns and would appreciate your feedback again. If you believe we overlooked any issues, please let us know.

Our kind regards,

David Esteves

Please see the attachment for the images

Reviewer 2 Report

The work «Development of MWCNT/ magnetite flexible triboelectric sensors by magnetic patterning» is devoted actual and significant topic about application of materials based on polyurethane matrix of electronic devices and  reduction in electrical percolation threshold polymer composites. The abstract briefly describes the motivation and results of the study. In the introduction, the authors describe well the problem being solved and the proposed ways for this. All synthesis methods contain a detailed description of the operations performed. The results obtained are presented in the form of figures of good quality and informative. The revealed regularities have an explanation.  And I have some comments:

1. There are no links to figures 1 and 2, which complicates their perception.

2. Please evaluate after revision why is Figure 10 a in the manuscript? Two points can be simply described.

3. The revealed regularities have an explanation. But the article completely lacks the conclusion section.

4. Fix errors, for example, page 3 - "(Error! Reference source not found.)".

5. In the experimental procedure and discussion, the mass fraction of the filler in the matrix is 1%. When adding magnetite, its mass ratio to MWCNT is indicated. It is difficult to understand how much filler is in the polymer matrix. magnetite is included in 1 mass percent or it is added additionally, i.e. filler more than 1 mass%.

Author Response

Reviewer 2

The work «Development of MWCNT/ magnetite flexible triboelectric sensors by magnetic patterning» is devoted actual and significant topic about application of materials based on polyurethane matrix of electronic devices and reduction in electrical percolation threshold polymer composites. The abstract briefly describes the motivation and results of the study. In the introduction, the authors describe well the problem being solved and the proposed ways for this. All synthesis methods contain a detailed description of the operations performed. The results obtained are presented in the form of figures of good quality and informative. The revealed regularities have an explanation.  And I have some comments:

Dear Reviewer, we appreciate your time reading and providing feedback on our manuscript.

  1. There are no links to figures 1 and 2, which complicates their perception.

You are right. There was a formatting problem when the template of the document was changed. All the figures are linked with the text now.

  1. Please evaluate after revision why is Figure 10 a in the manuscript? Two points can be simply described.

Thanks for your suggestion. The image present in Figure 10 a) was removed, and the information was explained in the text.

  1. The revealed regularities have an explanation. But the article completely lacks the conclusion section.

According to the Instructions for authors of mdpi, the conclusion is not mandatory, just the discussion. That is why we did not add a topic for the conclusion. However, we took your suggestion, and we changed the chapters. In chapter 3 we add the Results and Discussion and in chapter 4 we added the conclusions.

“Discussion: Authors should discuss the results and how they can be interpreted from the perspective of previous studies and of the working hypotheses. The findings and their implications should be discussed in the broadest context possible, and the limitations of the work highlighted. Future research directions may also be mentioned. This section may be combined with Results.

Conclusions: This section is not mandatory but can be added to the manuscript if the discussion is unusually long or complex.”

  1. Fix errors, for example, page 3 - "(Error! Reference source not found.)".

You are right. There was a formatting problem when the template of the document was changed. All the figures are linked with the text now.

  1. In the experimental procedure and discussion, the mass fraction of the filler in the matrix is 1%. When adding magnetite, its mass ratio to MWCNT is indicated. It is difficult to understand how much filler is in the polymer matrix. magnetite is included in 1 mass percent or it is added additionally, i.e. filler more than 1 mass%.

The weight percentage of magnetite is added (additionally) to 1 wt% of MWCNT. For example, for the sample (1:1), 1 wt.% magnetite is added to 1 wt.% MWCNT, which means that the composite has 2 wt.% of fillers. For the sample (1:1/24), 0.04 wt.% magnetite is added to 1 wt.% MWCNT, which means that the composite has 1.04 wt.% of fillers.

The addition of magnetite to MWCNT was explained in the results of the electrical characterization; however, now it was also clarified in the experimental procedure.

Dear Reviewer, we are confident we have addressed all your concerns and would appreciate your feedback again. If you believe we overlooked any issues, please let us know.

Our kind regards,

David Esteves

Reviewer 3 Report

The manuscript “Development of MWCNT/magnetite flexible triboelectric sensors by magnetic patterning” is submitted to Polymers. The works deal with the fabrication of soft robotic sensors. The manuscript is interesting for future readers. I would recommend a minor revision. The following questions have to be answered before acceptance:

1)      Magnetic particles reduce the resistance of the composite. Why? It is not clear from the text

2)      Please check the References “Error” on pages 3, 5, etc. It is difficult to read the text with these “Errors”.

1)      Figure 3. The Viscosity has the unit of mPa.s. In the text is only Pa.s. Please, choose only one.

Author Response

Reviewer 3

The manuscript “Development of MWCNT/magnetite flexible triboelectric sensors by magnetic patterning” is submitted to Polymers. The works deal with the fabrication of soft robotic sensors. The manuscript is interesting for future readers. I would recommend a minor revision. The following questions have to be answered before acceptance:

Dear Reviewer, we appreciate your time reading and providing feedback on our manuscript.

  • Magnetic particles reduce the resistance of the composite. Why? It is not clear from the text

The influence of magnetite in the MWCNT (multi-walled carbon nanotube) composite is shown in Figure 10, which is explained in the paragraph below.

From the electrical resistance measurements, we concluded that the addition of 1 wt% of magnetite to a PU (polyurethane) matrix does not affect the electrical properties of PU, maintaining its electrical insulator properties. Conversely, adding even small quantities of magnetite to 1 wt% of MWCNT improves the electrical resistance of the composite when compared with a sample of 1 wt% of MWCNT.

We understand that magnetite particles have higher magnetic susceptibility than MWCNT, making them more easily attracted to permanent magnets. When both particles are mixed in a solution and a permanent magnet is placed beneath the mould, the magnetic field pulls the magnetite particles along with the MWCNT. Increasing the amount of magnetite in the sample, such as in the case of (1:1), where 1 wt% magnetite is added to 1 wt% MWCNT, improves the pattern quality. This means that more MWCNT can be present in the electrode locations, reducing the electric resistance of the magnets.

Conversely, using a smaller amount of magnetite, such as adding 0.04 wt% magnetite to 1 wt% MWCNT (1:1/24), results in a smaller response of the particles to the permanent magnets. With reduced responsiveness to the permanent magnets, the magnetic pattern contains fewer MWCNT particles, decreasing the amount of connections between MWCNT particles at the electrode locations. By decreasing the amount of connection between MWCNT particles, the electrical resistance of the composite increases since there are few paths for the electrons to move.

 “ A decrease in the weight percentage of magnetite is observed to increase the electrical resistance of the composites, although keeping them electrically conductive. The average electrical resistance measured for the electrodes of (1:1) composite (1 wt.% MWCNT + 1 wt.% magnetite) was (5.31x103 ± 0.29x103) Ω, while for the composite with 1 wt.% MWCNT + 0.04 wt.% magnetite (1:1/24) it increased to (29.2x103 ± 10.1x103) Ω. Therefore, a gradual decrease in the magnetite content causes a gradual increase in both the electrical resistance and its standard deviation. Even though the composite with 1 wt.% magnetite is nonconductive, adding even a small amount of magnetite to MWCNT significantly decreases the electrical resistance and reproducibility of the fabrication process. The increase in the standard deviation of electrical resistance as the concentration of magnetite decreases is probably related to a decrease in the effectiveness of the “magnetic net” formed between MWCNT and magnetite in the composite. Decreasing the concentration of magnetite will hamper the continuous and homogeneous drag of the MWCNT agglomerates towards the permanent magnets, decreasing the concentration of the electrically conductive particles in their vicinity.”

2)      Please check the References “Error” on pages 3, 5, etc. It is difficult to read the text with these “Errors”.

You are right; there was a formatting problem when the document’s template was changed. All the figures are linked with the text now.

1)      Figure 3. The Viscosity has the unit of mPa.s. In the text is only Pa.s. Please, choose only one.

The viscosity units were changed to mPa.s in the whole text.

Dear Reviewer, we are confident we have addressed all your concerns and would appreciate your feedback again. If you believe we overlooked any issues, please let us know.

Our kind regards,

David Esteves
